# Dicer-like 5 deficiency confers temperature-sensitive male sterility in maize

Chong Teng [1,5], Han Zhang[2,5], Reza Hammond[3], Kun Huang[3], Blake C. Meyers [1,4 ✉] & Virginia Walbot [2 ✉]

Small RNAs play important roles during plant development by regulating transcript levels of target mRNAs, maintaining genome integrity, and reinforcing DNA methylation. *Dicer-like 5* (*Dcl5*) is proposed to be responsible for precise slicing in many monocots to generate diverse 24-nt phased, secondary small interfering RNAs (phasiRNAs), which are exceptionally abundant in meiotic anthers of diverse flowering plants. The importance and functions of these phasiRNAs remain unclear. Here, we characterized several mutants of *dcl5*, including alleles generated by the clustered regularly interspaced short palindromic repeats (CRISPR)–*Cas9* system and a transposon-disrupted allele. We report that *dcl5* mutants have few or no 24-nt phasiRNAs, develop short anthers with defective tapetal cells, and exhibit temperature-sensitive male fertility. We propose that DCL5 and 24-nt phasiRNAs are critical for fertility under growth regimes for optimal yield.

[1] Donald Danforth Plant Science Center, St. Louis, MO 63132, USA. [2] Department of Biology, Stanford University, Stanford, CA 94305, USA. [3] Delaware Biotechnology Institute, University of Delaware, Newark, DE 19716, USA. [4] Division of Plant Sciences, University of Missouri—Columbia, Columbia, MO 65211, USA. [5] These authors contributed equally: Chong Teng, Han Zhang. ✉email: bmeyers@danforthcenter.org; walbot@stanford.edu

Three major classes of endogenous small RNAs (sRNAs) exist in plants: microRNAs (miRNAs), heterochromatic small interfering RNAs (hc-siRNAs), and phased, secondary small interfering RNAs (phasiRNAs). From extensive sRNA sequencing in plants, numerous loci generating phasiRNAs have been reported; in grasses, phasiRNAs are enriched in flowers, particularly male reproductive organs[1–5]. In this context, phasiRNA production is initiated by miRNA-mediated cleavage of RNA polymerase II transcripts of two classes of *PHAS* loci. Subsequently, the 3′ portion of cleaved transcripts is converted to double-stranded RNA, a substrate for precise chopping by DICER-LIKE 4 (DCL4) yielding 21-nt products; a distinct, proposed role for DCL5 is generation of 24-nt phasiRNAs[3–5]. The 21-nt phasiRNAs are highly abundant during initial cell fate setting in maize anthers[1] and are important for male fertility in rice[6,7]. The 24-nt phasiRNAs accumulate coincident with meiotic start, peak during meiosis, and persist at lower levels afterwards; this pattern has generated speculation that they regulate meiosis[8,9].

In angiosperms, five DICER-LIKE (DCL) proteins have been described and partially characterized[10]. DCL1 is important for miRNA biogenesis, as illustrated by regulation of meristem determinacy in maize inflorescences[11]. In *Arabidopsis thaliana*, DCL2 processes viral-derived and transgene-derived siRNAs[12]; DCL3 produces 24-nt hc-siRNAs, which then direct DNA methylation of target loci[13]. In many plants including *Oryza sativa* (rice), DCL4 generates 21-nt *trans*-acting siRNAs and phasiRNAs[4]. Although the functions of these four *DCL* genes are well conserved in flowering plants, these genes have partially overlapping functions[12,14]. A fifth, more recently described and named gene, *Dcl5*, is monocot-specific[10]. To better understand the function of DCL5 and its proposed role in 24-nt phasiRNA biogenesis[1], we characterized maize *dcl5* mutants generated using either a high efficiency CRISPR-mediated gene editing system[15] or transposon-based mutagenesis. Here, we report that *dcl5* mutants have few or no 24-nt phasiRNAs, exhibit defects in tapetal cell differentiation, and show temperature-sensitive male fertility. We propose that DCL5-mediated generation of 24-nt phasiRNAs is required for fertility at temperatures that are optimal for maize yield.

## Results

**Null mutants of the maize *Dcl5* gene confer male-sterility.** Using a CRISPR system targeting the middle of the encoded protein[15], we generated four alleles for analysis (Fig. 1a), and from $T_0$ plants we developed stable lines that show Mendelian inheritance of *dcl5-1* (Fig. 1b, Supplementary Table 1 and Supplementary Fig. 1). *dcl5-1* and *dcl5-4* are frameshift mutants with transcript levels reduced to about a third of their wild type siblings; by contrast, *dcl5-2* (3 bp deletion) and *dcl5-3* (12 bp deletion, 1 bp substitution) have similar or higher transcript levels compared to control siblings (Fig. 1b and Supplementary Fig. 2). Wild type and heterozygous *dcl5-1//Dcl5* plants greenhouse-grown at 32 °C maximum day/21 °C minimum night were identical in whole plant architecture, anther morphology, and fertility. Under the same conditions, homozygous *dcl5-1* plants were male sterile (Fig. 1c). Compared to their fertile siblings, sterile *dcl5-1* plants lacked visible differences in tassels and spikelets, however, the sterile anthers were shorter, contained shrunken pollen, and did not exert from the spikelets (Fig. 1c and Supplementary Fig. 3). Sporadically, a few anthers exerted and shed viable pollen. Under field conditions, all four alleles showed male sterility and we did not observe any differences in phenotype between alleles. No genomic editing was detected in the homologous sequence in *Dcl3* which is the most likely potentially off-target locus

(Supplementary Fig. 4). To further, independently verify the observation of male sterility, we acquired an additional allele from the transposon-based Illumina-Mu population[16] (*dcl5-mu03*, line mu-illumina_139042.7). *dcl5-mu03* showed obvious male sterility in both greenhouse and field conditions (Supplementary Fig. 5 and Supplementary Table 1). We concluded that *Dcl5* is required for robust male fertility.

**dcl5-1 mutants exhibit tapetal defects.** To investigate whether *dcl5-1* plants display defects in anther cell patterning, we used confocal microscopy and found normal somatic layer architecture without defects in initial cell differentiation (Supplementary Fig. 6). In addition, chromosome pairing, alignment, and meiotic progression were normal in *dcl5-1* meiocytes (Supplementary Fig. 7a–l). The *dcl5-mu03* meiocytes also routinely completed meiosis and produced haploid gametophytes (Supplementary Fig. 7m, n). Transmission electron microscopy was utilized to visualize nuclei and other cell organelles at higher resolution. During the mid-meiosis stage (2–2.5 mm anthers), normal tapetal cells were densely packed with dark-staining materials and were mostly binucleate (Fig. 2a and Supplementary Fig. 8). By contrast, the tapetal cells in *dcl5-1* were pale and mostly mononucleate, and many cell organelles were not clearly resolved (Fig. 2b and Supplementary Fig. 8). Whole mount, fluorescence microscopy was used to quantify binucleate status in *dcl5-1* and fertile anthers. In 1.5 mm anthers (prophase I), there were five-fold fewer binucleated tapetal cells in *dcl5-1*; a significant difference persisted at 2.5 mm (meiosis II) and a large difference remained at 3.0 mm (post-meiosis) (Fig. 2 and Supplementary Fig. 9). In extensive confocal microscopic examination of *dcl5-1* anthers, we observed no significant or consistent differences in cell number or volume compared to fertile siblings. About one week after the completion of meiosis (2.5 mm stage), the tapetum starts programmed cell death, vacuolates, collapses, and disappears in the *dcl5-1//Dcl5* anthers (Supplementary Fig. 10); by contrast, the tapetum is developmentally delayed and retained in the *dcl5-1* anthers (Supplementary Fig. 10). The middle layer in *dcl5-1//Dcl5* collapses and disappears, yet it is largely retained in *dcl5-1* (Supplementary Fig. 10). Collectively, these observations indicate that tapetal development is delayed or arrested in *dcl5* mutants and that the *dcl5* tapetal cells are therefore likely defective in conducting post-meiotic functions supporting pollen maturation.

**Tapetum-specific and meiocyte-specific *Dcl5* expression.** To obtain a comprehensive spatiotemporal profile of maize *Dcl5* expression, we queried microarray, published and newly-generated RNA-seq (Supplementary Table 2) and proteomics data. *Dcl5* transcripts are highly enriched in tassels, cobs, embryos, and seeds[17,18], as well as fertile anthers. By contrast, DCL5 protein is low in 1.0 mm, pre-meiotic anthers and extremely high in 2.0 mm mid-meiosis anthers, but is undetectable in ear, embryo, or endosperm[19]. Microarray analysis of laser-microdissected anther cell types[20,21] and in situ hybridization analysis of anther lobes[1,22] established that *Dcl5* transcripts are highly enriched in the tapetum and are present at lower levels in pre-meiotic pollen mother cells and meiocytes. RNA-seq confirmed *Dcl5* expression in isolated maize meiocytes, however, quantitatively there are even higher levels in whole anthers[23], as confirmed by newly-generated anther data (Supplementary Table 2). Therefore, *Dcl5* is expressed much more highly in one or more somatic cell types than in meiocytes[23]. Integrating prior observations that 24-nt phasiRNA biogenesis is contingent on a normal tapetum[1], with the peak of *Dcl5* and DCL5 expression in meiotic anthers, we conclude that the 24-nt phasiRNA pathway is localized in tapetal cells. We hypothesize that this localization and

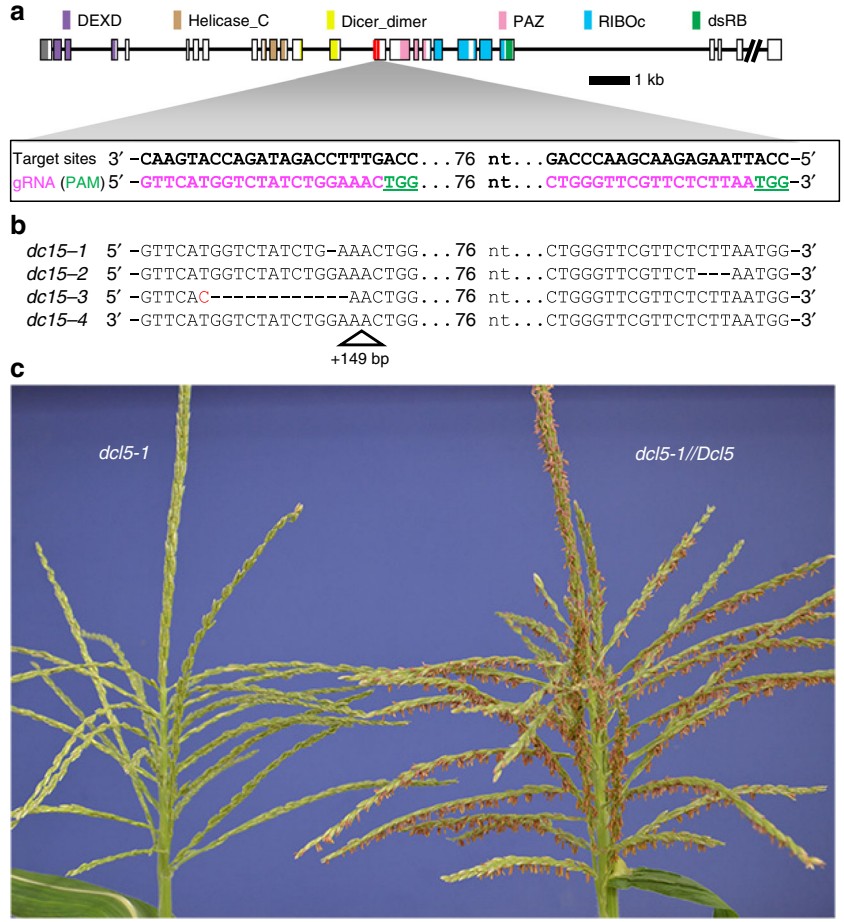

**Fig. 1 CRISPR-Cas9 was used in maize to mutate *Dcl5*. a** Schematic diagram of the *Dcl5* gene model. Conserved domains were identified using the NCBI conserved domain search tool against database CDD v3.16. All domains were found with extremely high confidence except the double stranded RNA-binding (dsRB) domain, which had an e-value of 0.05. **b** Sequences of the four CRISPR-Cas9-generated mutant alleles selected for this study. **c** A *dcl5-1* tassel lacking any exerted anthers (left) and a sibling heterozygous *dcl5-1//Dcl5* plant at peak pollen shed with hundreds of exerted anthers (right).

timing is of functional importance for the redifferentiation of tapetal cells into secretory cells during the meiotic period.

**24-nt phasiRNA abundance is greatly reduced in *dcl5* mutants.** To investigate if 24-nt phasiRNA biogenesis is affected in *dcl5* mutants, sRNA libraries were constructed from anthers or spikelets of each *dcl5* allele (Supplementary Table 2). Previously, we found that 24-nt phasiRNAs were readily detected from 176 24-*PHAS* loci in W23 inbred anthers[1]. The 24-nt sRNAs produced from all these loci were reduced dramatically in plants homozygous for each *dcl5* allele compared to fertile siblings even though *dcl5-2* and *dcl5-3* encode proteins that lack only a few amino acids (Fig. 3 and Supplementary Fig. 5c). Other sRNAs, including 24-nt hc-siRNAs and 21-nt phasiRNAs were retained (Supplementary Fig. 11). We conclude that *Dcl5* is required for 24-nt phasiRNA biogenesis and that the *dcl5-2* and *dcl5-3* mutations define amino acids essential for DCL5 function. Analysis of 24-PHAS precursors in RNA-seq anther libraries from the same plants showed greater or nearly the same levels of precursor abundance in each of the *dcl5* mutant alleles (Supplementary Fig. 12). Therefore, the absence of functional DCL5 severely disrupts processing of 24-nt phasiRNA precursors with a modest impact on their accumulation.

Analysis of these RNA-seq data to characterize downstream transcriptional pathways impacted in *dcl5* mutants was uninformative, consistent with prior results that failed to identify

mRNA targets for 24-nt phasiRNAs[1]. The transcriptional changes in *dcl5-1* mutants were minimal (23 transcripts altered two-fold) and mainly corresponded to genes expressed in later tapetal stages (Supplementary Table 3). Alterations in the transcript population are likely indirect and reflective of defective tapetal development.

***dcl5* mutants are temperature-sensitive male sterile.** During summer, 2016, *dcl5-1* field-grown plants exhibited variable fertility under typical summer conditions in Stanford, CA: 25/18 °C interrupted by multi-day heat waves exceeding 32 °C and short periods of cooler weather. To test whether temperature is a restrictive condition, three controlled greenhouse regimes were chosen — 28/22 °C, 26/20 °C, and 23/20 °C (14 h day/10 h night temperature) — and maintained within narrow limits over the life cycle (Fig. 4 and Supplementary Fig. 13). At 28/22 °C, *dcl5-1* mutants had shorter anthers that never exerted, while heterozygous siblings developed normally. By contrast, under both the 26/20 °C and 23/20 °C regimes, *dcl5-1* mutants were partially to fully fertile (see "Methods" section for details of tassel development and Supplementary Fig. 13 for quantification of anther emergence); this conditional male sterility was also observed in the transposon-disrupted *dcl5-mu03* allele (Supplementary Fig. 14). Under permissive conditions, *dcl5-1* pollen was viable based on Alexander staining despite a prolonged life cycle and delayed flowering date under cooler temperatures

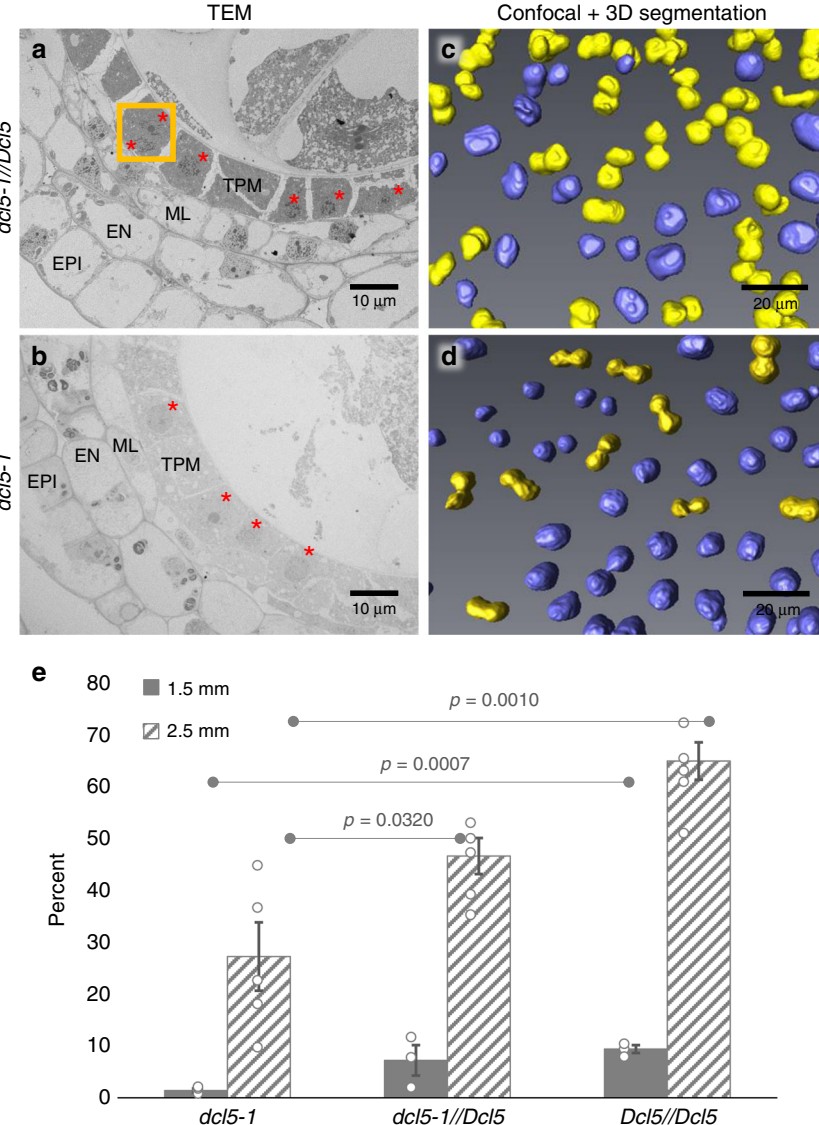

**Fig. 2 Tapetal cells in *dcl5-1* plants are delayed or arrested in achieving binucleate status. a** Transmission electron microscopy (TEM) of a fertile *dcl5-1// Dcl5* 2.0 mm anther lobe. At this stage, the tapetal cells (TPM) are packed with dark staining materials, likely exine components later secreted onto haploid microspores. The middle layer (ML), endothecial (EN), and epidermal (EPI) cells are highly vacuolated. The orange boxes mark two binucleated tapetal cells in which both nuclei were visible (red dots). **b** TEM of a sterile *dcl5-1* anther lobe at the same stage, demonstrating distended, pale-staining, and mononucleate tapetal cells. Two anthers as replicates from *dcl5-1//Dcl5* and two anthers from *dcl5-1* were used for TEM study. The other replicate is shown in Supplementary Fig. 8. **c** *dcl5-1//Dcl5*, and **d** *dcl5-1:* three-dimensional reconstructions of cleared, 2.5 mm anthers stained with the nuclear marker Syto13. Individual nuclei of mononucleate tapetal cells are marked in purple, and binucleated cells are marked in yellow. **e** Quantification of binucleated cells in *dcl5-1*, *dcl5-1//Dcl5*, and *Dcl5//Dcl5* siblings at the meiotic I (1.5 mm) and meiotic II (2.5 mm) stages, in a family segregating 1:2:1. Statistics: two-tailed Student's *T*-test. Three replicates were used for 1.5 mm anthers of each genotype; five replicates were used for 2.5 mm anthers of each genotype. Gray dots represent individual replicates (see Source Data File). Error bars indicate standard error of the mean (s.e.m.).

(Supplementary Fig. 15); importantly, ears pollinated by such pollen yielded viable progeny seed. At least one aspect of normal tapetal development was also restored: the percentage of binucleated tapetal cells at 1.5 and 2.5 mm in *dcl5-1* anthers was similar to their normal siblings (Supplementary Fig. 16). By contrast, in comparing *dcl5-1* transcript levels, 24-nt phasiRNA production, and *PHAS* precursor accumulation in *dcl5-1* plants under restrictive and permissive temperatures, we observed no significant differences; fertility-restored plants did not accumulate 24-nt phasiRNAs (Fig. 3, Supplementary Figs. 2 and 12). Our interpretation is that the functions of *Dcl5* and 24-nt phasiRNAs are dispensable for tapetum development and maize male fertility at low temperatures (23/20 °C) but are required at higher

temperatures. Cool temperatures slow the pace of development and may allow alternative pathways independent of the 24-nt phasiRNAs to support tapetal cell redifferentiation.

Maize tassels contain anthers representing seven days of development[24], and we wondered if sporadic anther exertion in *dcl5-1* plants reflected a short phenocritical period when some anthers experienced permissive conditions. Homozygous *dcl5-1* plants were greenhouse-grown at 28/22 °C until the tassel inflorescence formed (~30 days), then plants were moved into two walk-in chambers: permissive 23/20 °C or restrictive 28/22 °C regimes. In the next three weeks, 14 sets of three plants were swapped between the two regimes for 3, 6, 9, 12, 15, 18, or 21 days, and then all plants finished their life cycle in the 28/22 °C

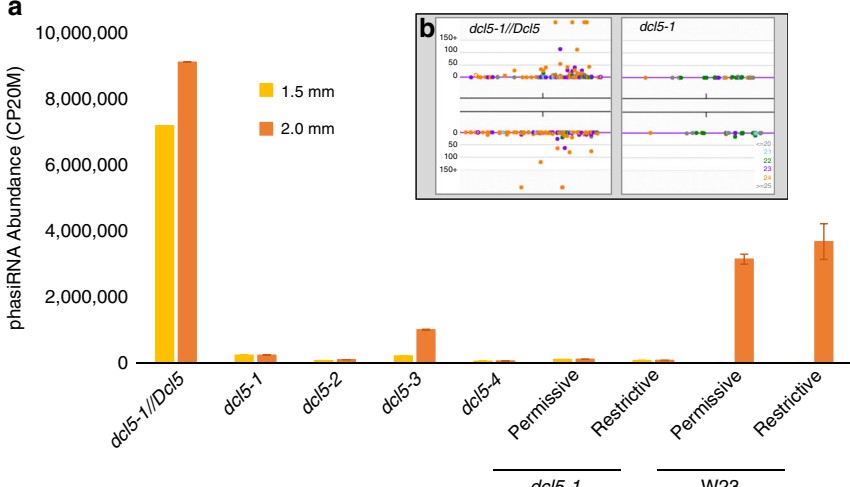

**Fig. 3 Near absence of 24-nt phasiRNAs in CRISPR-derived *dcl5* mutant alleles. a** Total 24-nt phasiRNAs from 176 *24-PHAS* loci are highly abundant in both 1.5 and 2.0 mm fertile heterozygous anthers, but abundance is reduced by >99% in *dcl5-1, dcl5-2,* and *dcl5-4,* and by >90% in *dcl5-3* anthers. Altered temperature regimes do not restore 24-nt phasiRNA abundances in either fertile (permissive) or sterile (restrictive) *dcl5-1* anthers (see main text). W23 inbred fertile anthers have similar 24-nt phasiRNA abundance under both temperature regimes. Error bars indicate standard error of the mean (s.e.m.). **b**, **inset** Representative *24-PHAS* locus #12 (B73, v4; chr 1, 178454619 bp) in a genome browser showing ~1.5 kbp with sRNAs in the fertile heterozygote (left) and *dcl5-1* (right). Orange dots are individual 24-nt phasiRNAs; *x*-axis is genome position on the top or bottom strand, and the *y*-axis depicts abundance in each genotype. Other colored dots are as indicated in the key, lower right.

greenhouse (Supplementary Fig. 17). Plants in the 23/20 °C regime from the start of meiosis through the release of mononucleate microspores (~9 day period, swaps 7–11, Supplementary Fig. 17) were more fertile. Sample plants were assessed during the 6th through 9th days to check anther staging in the main tassel spike, confirming that this interval corresponds to meiosis and post-meiotic stages in the most mature part of the tassel. These stages also encompass the initiation, peak, and continued presence of 24-nt phasiRNAs in normal anthers[1]. Plants in the permissive temperature for fewer than nine days during this interval had reduced fertility (swaps 1–6, 12–14); we hypothesize that this extended interval is required to allow all anthers to proceed through the phenocritical period to ensure full fertility. The temperature swap experiment demonstrates that there is a short period in which DCL5 is required to "buffer" development at elevated temperatures. Surprisingly, the restrictive temperature is similar to what is considered optimal day temperature for the U.S. corn belt; historical data indicate that growth between 20 and 29 °C generates optimal yield, with higher yields associated with warmer temperatures within this range[25,26].

## Discussion

Maize DCL5 is essential for robust male fertility: *dcl5* plants are temperature-sensitive during meiosis, show arrested tapetal development at this and later stages, and lack 24-nt phasiRNAs. In rice, perturbed 21-nt, premeiotic phasiRNAs confer photoperiod and temperature-sensitive male fertility[27–29]. Molecular and in situ analyses in maize indicate that the 24-nt phasiRNA biogenesis factors and these sRNAs are located primarily in the tapetum[1]. For maize, we conclude that 24-nt phasiRNAs play a direct role in tapetal redifferentiation and only an indirect role in meiocytes or gametophytes. We hypothesize that the 24-nt phasiRNAs are essential for tapetal redifferentiation for biosynthesis and secretion of materials that support haploid microgametophytes. The molecular mechanisms by which 21- and 24-nt reproductive phasiRNAs act remain unclear, because they lack complementarity to mRNAs.

Conditional male sterility is common in rice. A number of loci regulating photoperiod-sensitive genic male sterility (PMS) and thermo-sensitive genic male sterility (TMS) have been mapped[27,28]. Two of these loci, *PMS1* and *PMS3*, encode long noncoding RNAs[28,29]. *PMS1* is a *PHAS* locus that generates 21-nt phasiRNAs associated with male fertility under long-day conditions[28]. *PMS3* also produces a small RNA of 21-nt[29]. The rice *rna-dependent rna polymerase 6-1* (*rdr6-1*) mutant displays various degrees of spikelet defects at high temperatures[5]. These findings suggest that long non-coding RNAs and small RNAs are important regulators of male fertility controlled by cross-talk between genetic pathways and environmental factors. Abiotic stress-induced male sterility has been associated with alterations in tapetal development in many plant species[30]. For example, *TMS10* and its close homolog *TMS10L* encode two rice receptor-like kinases and are essential for tapetal degeneration and male fertility under high temperatures[31].

In maize, conditional or partial male sterility has been reported, but not extensively studied. The maize *ocl4-1* and *ocl4-2* mutants both show varying degrees of male sterility, and the severity of the phenotype is strongly influenced by environmental conditions[32]. OCL4 is a HD-ZIP IV transcription factor[32], and *ocl4* mutants lack 21-phasiRNAs[1]. The OCL4 regulation of 21-nt phasiRNA biogenesis and the fact that male sterility in the *dcl5* mutants is also sensitive to temperature indicate that the functions of some secondary small RNAs support normal development and cellular functions during environmental fluctuations outside the optimal temperature regime. The finding that *dcl5-1* mutants are sterile under the 28/22 °C regime suggests that 24-nt phasiRNA biogenesis is essential for high corn yield under optimal U.S. summer farming conditions.

We can only speculate why the incredibly diverse and abundant 24-nt phasiRNAs are required to support normal tapetal development under mild temperature stress. Plants were well-watered under all growth conditions, and there were no obvious indicators of water stress such as leaf rolling or leaf damage. Under warm temperatures, plant development is accelerated relative to cooler conditions: the transition to flowering is earlier and pollen shed is precocious. In the case of the temperature

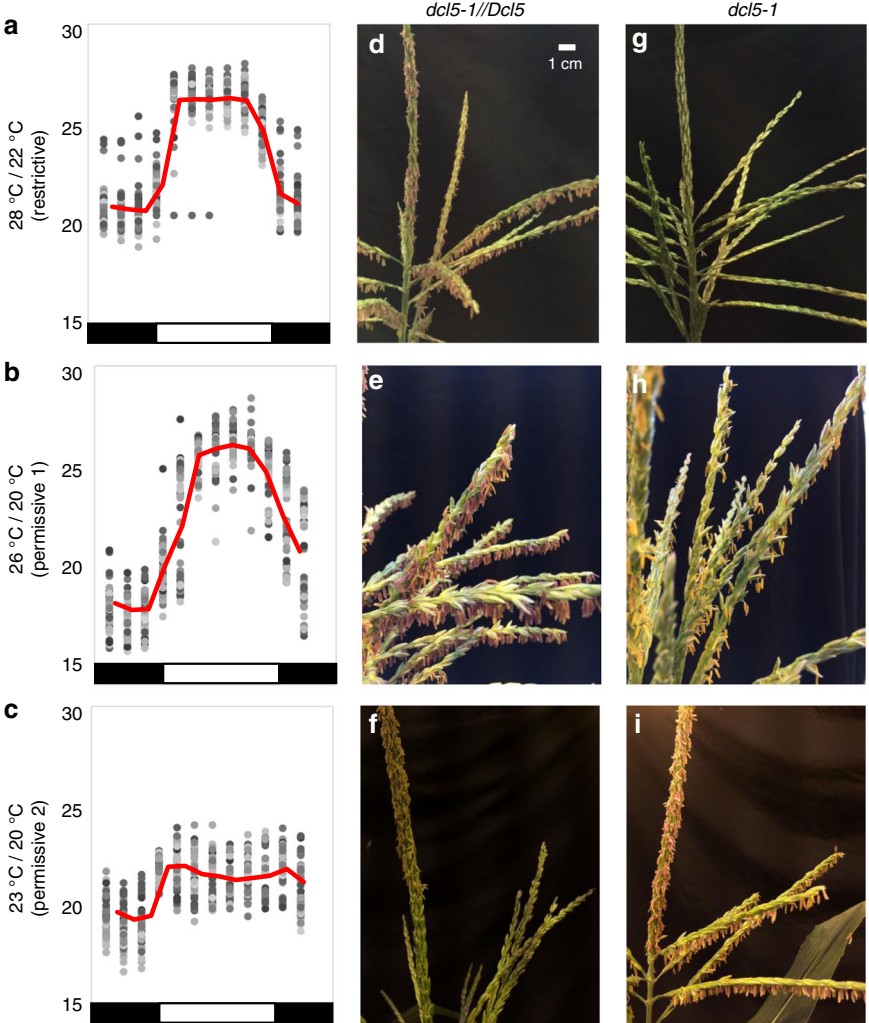

**Fig. 4 _dcl5-1_ anther fertility is temperature sensitive.** Three sets of _dcl5-1_ and sibling _dcl5-1//Dcl5_ plants grown in greenhouses with differing temperature regimes: 28/22 °C, 26/20 °C, or 23/20 °C (day/night). **a**–**c** Temperatures were recorded every 2 h for 60 days (gray dots, average in red; day/night indicated as white or black, below). **d**–**f** Heterozygous _dcl5-1//Dcl5_ siblings were fully fertile under all three regimes. **g**–**i** _dcl5-1_ plants in the restrictive regime (**g**) were completely male sterile, while those in permissive conditions were partially (**h**) or fully (**i**) fertile. Images were taken at 60 ± 3 days after planting (DAP) (top row), 70 ± 2 DAP (middle row), and 71 ± 3 DAP (bottom row); the pace of tassel development is temperature-dependent and full anther exertion occurs at different days after planting, for example **f** was photographed on the first day of anther exertion, a seven-day process (see "Methods" section). The scale bar in **d** is approximate and pertains to all tassel images.

regimes we employed, pollen shed was about 10 days earlier under the 28/22 °C growth condition than under cooler daytime temperatures (Fig. 4). It is possible that during more rapid growth, there is a more stringent requirement for coordination of developmental processes, requiring more rapid and complete switches from one developmental state to another. Tapetal cells redifferentiate during meiosis, evidenced by accumulation of future exine components and binucleate status. Because this is the only anther cell type with obvious defects at peak 24-nt phasiRNA production at 2.0 mm, we hypothesize that the abundant phasiRNAs are used to reset tapetal capabilities. Because 24-nt phasiRNAs lack a high degree of complementarity to mRNAs other than their own precursors, one hypothesis is that it is their sheer abundance that permits a resetting of development. For example, the 24-nt phasiRNAs could bind to diverse AGO proteins, in effect displacing miRNAs or other sRNAs involved in gene regulation associated with pre-meiotic tapetal development. The 24-nt phasiRNAs could thus contribute to rapid redifferentiation of tapetal cells during meiosis by eliminating gene expression regulation persisting from an earlier developmental

state. It is also possible that 24-nt phasiRNAs perform this or another role in meiocytes or in other anther cell types, although their abundance is much less than in non-tapetal cells. Alternatively, the functions of 24-nt phasiRNAs may involve targeted action utilizing Argonautes or other RNA-binding proteins with as-yet uncharacterized activities required in tapetal differentiation.

Our study in maize establishes that DCL5 plays a unique role, having subfunctionalized from DCL3 for 24-nt phasiRNA biogenesis in anthers. _Dcl5_ transcripts are still expressed in diverse organs, however, protein and critical functions are restricted to anthers. Expression analysis found just 23 genes differentially expressed in our _dcl5-1_ mutants, but all of these are likely reflective of arrested tapetal development. That the thousands of 24-nt phasiRNAs fail to impact mRNA abundances significantly indicates that 24-nt phasiRNAs are distinct from tasiRNAs which have specific mRNA targets and modulate transcript abundances. DCL5 and therefore the 24-nt phasiRNAs are dispensable at cool growing temperatures for maize, however, DCL5 is required for robust fertility at temperatures for optimal yield in the U.S. corn

belt. As climate disruption, including more extensive heat waves, and a general warming trend increase field temperatures, buffering maize tapetal development from adverse effects may be possible by enhancing 24-nt phasiRNA functions. The 24-nt phasiRNAs may permit development during other untested environmental challenges. Given the high degree of variation typically observed across maize inbreds for many traits, the *dcl5* mutant phenotype may also vary across inbreds. More generally, it is unknown how many crop plants utilize reproductive phasiRNAs and whether their tapetal health and functions could benefit by introduction or optimization of these or other sRNA pathways[30–32].

## Methods

***Dcl5* mutant generation and characterization**. To increase the CRISPR-based mutagenesis frequency in the targeted region, two target sites separated by 76 bp were selected for guide RNA (gRNA) construction, designed using the CRISPR Genome Analysis Tool[15]. Rice *U6* small nuclear RNA gene promoters (P*U6.1* and P*U6.2*) were used to drive the expression of each gRNA. Two guide RNAs (first gRNA, 5′-GTTCATGGTCTATCTGGAAAC-3′; second gRNA, 5′-CTGGGTTCG TTCTCTTAA-3′) were first selected; then two complementary oligonucleotides for each guide RNA (g*ZmDcl5*-F1, 5′-tgttGTTCATGGTCTATCTGGAAAC-3′; g*ZmDcl5*-R1, 5′-aaacGTTTCCAGATAGACCATGAAC-3′; g*ZmDcl5*-F3, 5′-gtgtGC TGGGTTCGTTCTCTTAA-3′; g*ZmDcl5*-R3, 5′-aaacTTAAGAGAACGAACCCA GC-3′) were annealed to produce double-stranded DNA oligos, then inserted into a pENTR based vector using *Btg*ZI and *Bsa*I (New England BioLabs) restriction enzymes. These constructs were transferred using Gateway recombination to the binary vector containing the *Cas9* gene driven by the maize *Ubiquitin 1* promoter and the *bar* gene driven by *35S* promoter[15]. The final construct was transformed into *Agrobacterium tumefaciens* for delivery into HiII maize immature embryos, followed by screening of positive calli with T7E1 assay and Sanger sequencing[15].

DNA samples were extracted from Basta®-resistant calli and screened for mutations in the target region by PCR using primers *Dcl5*_F1 (5′-ATCTAGATC TCCAGACCATTGAACCCTGTC-3′) and *Dcl5*_R1 (5′-GTATTCTAGACTTGA ATAACCTGCTCTTTG-3′). Positive calli were transferred to regeneration media; multiple plants regenerated from the same callus were considered to be biological clones[15]. No chimeric plants were detected, consistent with Cas9 action in a single cell. Leaves from mature T0 plants were genotyped for a second time using PCR amplification and Sanger sequencing to confirm the mutations. Over several generations families were established that segregated 1:1 for *dcl5*//+ fertile: *dcl5*// *dcl5* homozygous sterile individuals using standard maize genetic procedures.

*Mu* transposon population were first screened online (www.maizegdb.org and teosinte.uoregon.org). Three candidates from Illumina-Mu population were identified (*dcl5-mu01*, mu-illumina_252772.6; *dcl5-mu02*, mu-illumina_254833.6; *dcl5-mu03*, mu-illumina_139042.7); one candidate from UniformMu was identified (*dcl5-mu04*, UFMu-03512). The mutant information provided online showed the Mu insertions are expected in either 5′-UTR region (*dcl5-mu02* and *dcl5-mu04*) or introns (*dcl5-mu01* and *dcl5-mu03*), but we amplified the approximate region and found the third exon of *Dcl5* is disrupted in *dcl5-mu03* with Sanger sequencing, using primer *Dcl5*-F2 (5′-CATTAGTCAGGCAGTTGGTAGG-3′) and *Dcl5*-R2 (5′-GTGGCCCGTTTGT CTTCTCC-3′). Because the *Mu* transposon was in an exon, the *dcl5-mu03* allele was the most promising candidate and pursued further. The heterozygous plants were self-crossed to obtain a 3:1 segregating family, and homozygous plants were identified by PCR and used for molecular and phenotypic characterization.

**Microscopy and imaging**. For anther wall structure analysis with confocal microscopy, the anthers of precisely measured sizes were dissected and stored in 70% ethanol for fixation. Fixed anthers were then stained by propidium iodide and visualized on a Leica SP8 confocal microscope[20]. Meiocytes were extruded and stained by DAPI (4′,6-diamindino-2-phenylindole)[33]. The Alexander′s staining solution[34] was used to test the viability of the pollen grains in the permissive conditions. For TEM[35], anthers were dissected and fixed in fresh 0.1 M PIPES buffer (pH 6.8). A 2% osmium tetroxide stain was applied with brief washes, followed by dehydration in an acetone gradient. Specimens were infiltrated using a gradient of Spurr's resin (Sigma-Aldrich) and embedded. The sections were cut using a Leica UCT ultramicrotome, stained in uranyl and lead salts, and observed using a Zeiss TEM instrument with a LEO 912 AB energy filter. Semi-thin sections were obtained with the same method as TEM except for the heavy metal staining steps, and imaged with Aperio Digital Pathology Slide Scanner with a 40× lens.

**ScaleP clearing of maize anthers**. Maize anthers were fixed directly into 4% paraformaldehyde in phosphate-buffered saline (pH 7.4), and then prepared following the ScaleP protocol[36]. After 8 h KOH treatment, anthers were stained for a week at room temperature with 1 mM Calcofluor white for imaging of cell walls and 5 μM Syto13 (ThermoFisher) for imaging nuclei. Samples were then cleared for three days with ScaleP solution consisting of 6 M urea, 30% (v/v) glycerol, and 0.1% (v/v) Triton X-100 in sterile water. Cleared samples were then imaged with a Zeiss

880 multi photon confocal microscope with a 40× LD C-Apochromat water lens (numerical aperture of 1.1; working distance of 0.62 mm). The 3D rendering was performed with Amira (FEI, ThermoFisher) software. Tapetal cells were classified as mononucleate or binucleate manually for the 2-D images, or segmented automatically for the 3-D images, and then artificially colored.

**sRNA-seq and RNA-seq library construction and sequencing**. Total RNA for sRNA-seq and RNA-seq libraries was isolated using the PureLink Plant RNA Reagent (ThermoFisher). Total RNA quality was assessed by denaturing agarose gel electrophoresis and quantified by the Qubit RNA BR Assay Kit (ThermoFisher). For library preparation, 20 to 30 nt RNAs were excised from a 15% poly-acrylamide/urea gel, and ~25 ng of sRNA used for library construction with the TruSeq Small RNA Prep Kit (Illumina); alternatively, RealSeq®-AC kits (Soma-Genics) were used for ultra-low amount of total RNA starting from 100 ng. For RNA-seq, 2 μg of total RNA was treated with DNase I (New England BioLabs) and then cleaned with RNA Clean and Concentrator-5 (Zymo Research). The TruSeq Stranded Total RNA with RiboZero-Plant Kit (Illumina) or NEBNext Ultra II Directional RNA Library Prep Kit (New England BioLabs) was used for library construction with 500 ng of treated RNA. Sequencing in single-end mode on an Illumina HiSeq 2500 or NextSeq (University of Delaware) yielded 51 or 75 bp reads for both sRNA-seq and RNA-seq.

For sRNA-seq data[37], we first used Trimmomatic version 0.32 to remove the linker adapter sequences[38]. The trimmed reads were then mapped to version 4 of the B73 maize genome using Bowtie[39]. Read counts were normalized to 20 million to allow for the direct comparison across libraries. phasiRNAs were designated based on a 24-nt length and mapping coordinates within the previously identified 176 *24-PHAS* loci[1], updated to version 4 of the B73 genome using the assembly converter tool[40]. If a 24-nt phasiRNA did not uniquely map to the genome, we divided the abundance equally to each location to which the read mapped, i.e. a hits-normalized abundance. These hits-normalized abundances were then summed for each of the 176 loci to calculate the 24-nt phasiRNA abundances.

RNA-seq libraries were trimmed as above, and mapped to version 4 of the B73 genome using Tophat version 2.0.12 (ref.[41]). For differential expression analysis of genes, we assembled the transcripts using the Cufflinks package[42] and raw gene expression levels were quantified using FeatureCounts version 1.5.0 (ref.[43]) to generate count tables that were imported into R for statistical analysis[44]. Weakly expressed transcripts (fewer than 1 read) were filtered out, and the remaining transcripts were normalized using DEseq2 to identify a set of differentially expressed genes for anthers homozygous for each mutant allele[45]. Due to a lack of a replicate for the *dcl5-1* restrictive library, we calculated dispersions using the mean of libraries to identify differentially expressed transcripts without a complete set of replicates.

For differential expression analysis of the *PHAS* precursors, we were unable to assemble the transcripts that mapped to these regions because of the absence of annotation in any feature files. Instead, we conducted this analysis in parallel to the analysis of the sRNA data. We first normalized the mapped fragmented reads as counts per 20 million, and applied the hits-normalized abundance approach described above. Using the annotated *24-PHAS* loci (above), we then identified the summed abundance of RNA-seq transcript fragments that mapped to these regions to identify a representative abundance for each *24-PHAS* precursor. Replicate libraries were averaged together prior to the construction of the heatmap, and we added 1 to all abundances in order to prevent $\log_2(0)$ error while also giving the added benefit of removing negative log transformed values. Boxplots were generated for each replicate's log transformed values.

**Determination of male fertility**. Anther exertion on male fertile tassel take approximately seven days: Anthers emerge from the upper florets of the center of main spike on day 1; day 2 will include anthers from the lower florets in the center of the main spike and additional upper florets from the main spike (above and below the middle zone); day 3 will include lower floret spikes from the zones that stated on day 2, plus the upper florets at the base of the main spike and the middle of the primary branches; day 4 will include lower florets from the spikelets involved on day 3, plus upper florets from more regions on the primary branches; day 5 will include the lower florets from spikelets involved on day 4; day 6 will include the upper florets on the secondary branches, and day 7 will include the lower florets of the secondary branches[24]. Based on the maize anther emergence, tassel at similar stages in anther emergence were chosen for photographs and quantification.

Tassel RGB images were obtained with cell phones with digital cameras. Then, a quantitative image-based color trait analysis pipeline on the PlantCV v2 platform[46,47] (https://plantcv.danforthcenter.org) was applied to determine the male fertility, with a computational segmentation algorithm. In brief, the naïve Bayes approach was applied and three classes (anther; other tassel parts; and background) of labeled pixels were used to segment tassel images. By using these three classifiers, anthers and other parts of tassel were highlighted in red and green. Then, area of anther and tassel from each tassel was summed and used to calculate anther/tassel area ratio, as an indicator for male fertility.

**Reporting summary**. Further information on research design is available in the Nature Research Reporting Summary linked to this article.

## Data availability

Requests for materials should be addressed to walbot@stanford.edu. Raw data were submitted to GEO under GSE122449 (including sRNA-seq as GSM3466686 to GSM3466700 and GSM4180400 to GSM4180404; and RNA-seq data as GSM3466701 to GSM3466714, and GSM4180405 to GSM4180408), and the processed data are available via our maize genome browser at https://mpss.danforthcenter.org. Other supporting data are available from the corresponding authors upon request. The data underlying Figs. 2–4 as well as Supplementary Figs. 2, 5, 9, 11,12, 16, and 17 are provided as a Source Data file. Source data are provided with this paper.

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

## Acknowledgements

The *dcl5* CRISPR mutants were generously generated by Si Nian Char in Dr. Bing Yang's group at Iowa State University (now University of Missouri—Columbia) and were key to initiating the overall study. The *dcl5-mu03* allele was provided by Dr. Alice Barkan at the University of Oregon. TEM images were produced and imaged by Dr. Howard Berg at the Donald Danforth Plant Science Center. Semi-thin section images were produced with the help of Shannon Modla in the BioImaging Center at the University of Delaware. We thank the greenhouse staff at the Donald Danforth Plant Science Center for facilitating the temperature swap experiment, Sandra Mathioni for the preparation of small RNA and RNA-seq libraries, and Jeff Caplan for advice on microscopy. This work was supported by U.S. National Science Foundation Plant Genome Research Program (NSF-PGRP) awards 1649424 and 1754097. Author contributions: H.Z., B.C.M., and V.W. conceived of the project, H.Z., C.T., and K.H. performed and interpreted experiments, R. Z. analyzed sequencing data, H.Z. and C.T. wrote the manuscript with editing by V.W., B.C.M., and with input from all co-authors.

## Competing interests

The authors declare no competing interests.
