## [Peer Review File · Nature Communications]

Reviewers' comments:

Reviewer #1 (Remarks to the Author):

The exact functions of DCL5 and 24nt-phasiRNAs in reproductive organs of monocots have been a mystery since their discoveries. In this paper, the authors confirmed the involvement of DCL5 in 24-nt phasiRNA biogenesis and showed that DCL5 and 24-nt phasiRNAs play vital roles in anther developments only at high temperatures. The thermo-sensitivity and the tapetal cell defects of *dcl5-1* mutant are well described. In general, the manuscript is well written and the amount of work presented in the paper is appreciated.

A few major comments are listed below.

1. It is surprising to see that monocot plants make such efforts to cope with a possible condition of higher temperatures during reproduction. Why is this mechanism not evolved in dicot?
2. If DCL5 and 24-nt phasiRNAs are indispensable under higher temperature, is DCL5 expression or 24-nt phasiRNA production influenced by elevated temperature?
3. Given the fact that normal tetrad cells were observed in the *dcl5-1* mutant (Fig S4) but inviable pollens were seen later (Fig S2), it is interesting to determine what stage of gametogenesis is affected?
4. Related to the last point, the earliest phenotype reported here is the abnormal tapetal cells in 1.5 mm anther. Although the meiosis seems to be normal, suggesting that abnormality of tapetal cells may not affect meiosis, I would be very conservative to exclude the possibility that DCL5 and 24nt-sRNAs having function in meiosis.
5. It will be interesting to know that whether programmed cell death of tapetal cells is affected in *dcl5* mutant.
6. All tapetal cells would become binucleate at end of meiosis. Do slightly larger anthers (2.5-3.5mm) in *dcl5-1* mutant also have fewer binucleate tapetal cells? This data would clarify whether tapetal development delayed or arrested.
7. Line 76-78. "we conclude that the 24-nt phasiRNA pathway is localized in tapetal cells" might be over-stated if the precursor of 24-nt phasiRNA are considered to be part of the pathway. Since according to Zhai et al. (2015) 24-PHAS are present in tapetal cells and in meiocytes.

Minor points

1. Line 16: miRNA needs an abbreviation.
2. Fig S1 does not contain mRNA abundance data of *dcl5* alleles.
3. Fig S4, *dcl5-1* mutant meiocytes have more organelle? what are those tiny DAPI-stained dots?
4. Line 76, (4)?
5. Fig S7, *dcl5-1* mutant under permissive and restrictive temperature showed higher expression of non-phasi 24-nt sRNAs.?? What they are? Why it is not seen in the *dcl5-1* mutant at the 1.5 or 2 mm anthers?
6. Line 40-47: I suppose that *dcl5-2*, *dcl5-3* and *dcl5-4* plants are sterile, based on descriptions in lines 84-86. The authors should mention whether these three alleles are sterile or fertile with data shown in supplemental figures.
7. Figure A-E, G-K in Figure S4 should be enlarged for easier reading.
8. Both *dcl5-2* and *dcl5-3* allele have normal DCL5 expression but without 24nt phasiRNAs (Figure S1). It would be an interesting to examine whether any phenotypic difference can be observed in tapetal cells and meiosis.
9. Fig S5 is incomplete.

Reviewer #2 (Remarks to the Author):

The manuscript, entitled "Dicer-like 5 deficiency confers temperature-sensitive male sterility in maize" (Zhang, et al) reports novel findings regarding the role that DICER-LIKE 5 plays in maintaining maize male fertility. DICER-LIKE (DCL) proteins are essential for biogenesis of small regulatory RNAs in angiosperms. Unlike other members (DCL1, DCL2, DCL3, and DCL4) that are conserved across all flowering plants, DCL5 was recently found monocot-specific. In this manuscript, authors created maize *Dcl5* mutations using CRISPR-Cas9 gene editing technique, characterized maize *dcl5* mutants under different temperature-controlled growth condition and discovered that maize DCL5 is essential for robust male fertility. On the molecular level, this paper suggests that DCL5 corresponds to biogenesis of 24-nt phasiRNAs, and the phasiRNA pathway is believed localized in tapetal cells and plays important roles in plant reproductive process. Findings from this study has not only advanced our understanding of DCL5's role in maintaining a robust male fertility and responding to the temperature changes, but also has significance in agricultural practice. I only have some minor suggestion and comments.

1. By looking at the Figure 2, which seems to me that *dcl5-1* mutant anthers are not only developing less binucleate tapetal cells, but also abnormal in cell number and size. I am not sure if this is just from the particular section for imaging. From Figure 2B, it is clear that *dcl5-1* has smaller number but larger sized tapetal cells, middle layer cells, and endothelial layer cells—almost all the anther wall layers have larger sized and smaller numbered cells. Please clarify whether this applies to all *dcl5-1* anthers growing in 28°/22°C or in the field. Are there any statistic data regarding this abnormality.

2. This is just following my first question. If *dcl5-1* mutants develop fewer but larger anther wall cells, it suggests a role before meiosis and likely on anther cell differentiation. In this case, it would make sense that the accumulation of 24nt phasiRNA in meiocytes and anthers (mainly tapetal cells) are parallel processes. The reduced number of binucleate tapetal cells in *dcl5* mutant could be associated with less dynamic cell division. Are there any other changes that the authors have been notified such as changes of gene expression of cyclin or cyclin dependent kinase genes?

3. Since this paper suggests a correlation between the temperature sensitivity of DCL5 to the agricultural belt of North America, a couple more sentences to introduce the origin of maize Hi-II line would be helpful for readers. In addition, it may be helpful to mention that it may vary among maize inbreds, especially for those with tropical origin, such as CML228.

Reviewer #3 (Remarks to the Author):

Nature Communications
To Authors,

This study shows that DCL5 (which is DCL3b) is required for anther wall developments and this *dcl5* deficiency shows the temperature-sensitive male sterility through the decrease of 24-nt phasiRNAs in maize. However, it is already known that rice DCL3b, which is DCL5 in this paper, are required for 24-nt reproductive phasiRNA production in rice (Song et al., 2012). Furthermore, there are no correlation between 24-nt phasiRNAs expression and temperature-sensitive male sterility of *dcl5* mutants under restrictive and permissive temperature. It remains unknown the mechanism of temperature-sensitive male sterility via 24nt-phasiRNAs and the function of 24 reproductive phasiRNAs processed by DCL5. I would recommend against its publication in Nature Communications.

Major comments

1) DCL5 allele mutants

There are no data of mendelian inheritance of "EACH" dcl5 allele (page 2 Line37). Please show these results including under 28/22°C, permissive and restrictive temperature.

Please check whether there are other deletion or insertion regions in all mutants by whole genome sequences or other experiments, because there are 5 paralogues of DCLs in maize.

Please show the expression of Dcl5 in all mutants. I couldn't check the results of Fig S1.

2) Female effect in dcl5

How about female in dcl5 mutants?

3) Line49-63

Please explain the mono- and bi-nucleotide more detail with references.

4) Dcl5 expression and DCL5 protein expression (line69-79)

Please show DCL5 and DCL3a expression with in situ hybridization and show DCL5 and DCL3a protein localization in anther, because DCL3 is also important for 24-nt reproduction in dicots (Xia et al paper).

5) 24-PHAS expression

Please explain the difference of 24-PHAS expression in dcl5-4 and other three alleles (FigS8), though the expression of 24-nt phasiRNAs are decreased in all alleles (Fig3 and FigS7).

6) 21-nt phasiRNAs/21PHAS

Why were 21-nt phasiRNAs increased in all dcl5 mutants (FigS7) ?
How about 21PHAS expression in mutants?

7)

There are no significant differences of "24nt phasiRNAs expression (also 24-PHAS)" under restrictive and permissive temperature (Fig3 and FigS8). Authors concluded that 24-nt phasiRNAs are dispensable at cool growing temperature for maize and may allow alternative pathways (line111,112, 113150). Please show the evidence. For example, the authors performed RNA-seq using WT and dcl5 under restrictive and permissive temperature. How about changing of genes of alternative pathways independent of 24nt phasiRNAs to support the authors interpretation?

8) Targets of 24-nt phasiRNAs

The authors concluded that characterization of downstream transcription, target, was uninformative (line 92). Please show the results. Because it seems that premeiotic phasiRNAs (21nt?) are used for searching target, not meiotic phasiRNAs in Zhei et al (2015) paper, which are referenced (line 93).

9) It is recommended to fit this manuscript to Nature communications form.

Reviewers' comments:

Reviewer #1 (Remarks to the Author):

The exact functions of DCL5 and 24nt-phasiRNAs in reproductive organs of monocots have been a mystery since their discoveries. In this paper, the authors confirmed the involvement of DCL5 in 24-nt phasiRNA biogenesis and showed that DCL5 and 24-nt phasiRNAs play vital roles in anther developments only at high temperatures. The thermo-sensitivity and the tapetal cell defects of *dcl5-1* mutant are well described. In general, the manuscript is well written and the amount of work presented in the paper is appreciated.

We appreciate the positive comments.

A few major comments are listed below.

1. It is surprising to see that monocot plants make such efforts to cope with a possible condition of higher temperatures during reproduction. Why is this mechanism not evolved in dicot?

As described in a companion paper, this pathway is present in some but not all eudicots. It's possible that the absence of the 24-nt phasiRNAs makes anthers susceptible to higher temperature via an indirect effect that is not a reflection of the primary role of the 24-nt phasiRNAs. Until we know the precise molecular function of the 24-nt phasiRNAs, we can only speculate about the eudicot-monocot differences. Nonetheless, we agree with the reviewer that this difference is surprising.

2. If DCL5 and 24-nt phasiRNAs are indispensable under higher temperature, is DCL5 expression or 24-nt phasiRNA production influenced by elevated temperature?

We have not tested *dcl5* transcript levels under controlled temperature conditions, but have assayed plants grown in the greenhouse and field. This point was directly addressed, however, for the *dcl5-1* mutant anthers (Fig. S2).

For 24-PHAS precursors Fig. S10 reports that actual accumulation in *dcl5* under different temperature regimes for all four mutant alleles plus *dcl5-1* under permissive and restrictive temperatures (Fig. S10).

3. Given the fact that normal tetrad cells were observed in the *dcl5-1* mutant (Fig S4) but inviable pollens were seen later (Fig S2), it is interesting to determine what stage of gametogenesis is affected?

In this report we are focusing on the timing and cellular impact of the loss of *dcl5* coincident with the start of expression and as early as we can detect a phenotype. We agree that ultimately understanding the arrest stage of this, and indeed all other male-sterile mutants in which meiosis is normal, is of general interest, however the arrest of anther development is a terminal phenotype resulting from earlier "failures". Studying the terminal step sheds little light on the initial requirement for 24-nt phasiRNAs or on the direct consequences of their absence.

4. Related to the last point, the earliest phenotype reported here is the abnormal tapetal cells in 1.5 mm anther. Although the meiosis seems to be normal, suggesting that abnormality of tapetal cells may not affect meiosis, I would be very conservative to exclude the possibility that DCL5 and 24nt-sRNAs having function in meiosis.

There are two considerations here:

1. The impacts on the tapetum are obvious, and the tapetum is the primary site of 24-nt biogenesis. By *in situ* hybridization, we previously found (Zhai et al. 2015 PNAS) some evidence for 24-nt phasiRNAs in meiocytes, however, the signal was primarily over the callose coat. Probe adherence to carbohydrate is a common artifact in the method. In data now on bioRxiv (DOI: 10.1101/434993) by different members of our group, we have conducted single cell RNA-seq on many stages of PMC and meiocytes. We find no evidence for the biogenesis machinery for 24-nt phasiRNAs in these cells. These data, which we haven't cited since the manuscript is still under peer review, cannot exclude that 24-nt small RNAs are transferred to the meiotic cells.
2. A healthy tapetum is obviously important prior to meiosis for remodeling of the callose coat, and it is perhaps important during meiosis, although the callose coat should exclude entry of most materials into the meiocytes. The healthy tapetum is important, however, for tetrad release, exine formation, and nutrition of the microgametophytes at the completion of meiosis. That meiosis appears to be normal in *dcl5-1* mutant anthers indicates that neither the presence of 24-nt phasiRNAs in the tapetum nor their hypothetical transfer to the meiocytes are required for completion of meiosis. This does not rule out that the tapetum directly or indirectly "regulates" meiosis, and thus we don't mean to exclude the possibility that DCL5 and 24-phasiRNAs have functions in meiosis, but rather we are emphasizing the importance and requirement of DCL5 and 24-phasiRNAs in tapetal development. Earlier acting tapetal mutants such as *ms23*, *ms32* in maize, and others in Arabidopsis and rice fail during meiosis, suggesting that tapetal biology and possibly transfer of signals or materials to PMC is essential for meiosis (reviewed by Walbot and Egger 2016).

5. It will be interesting to know that whether programmed cell death of tapetal cells is affected in *dcl5* mutant.

We have not investigated this, as it is at a later stage than the focus of our work. The entire anther is failing at about the stage when PCD starts in the tapetum. Perhaps this could be investigated in a future study. Our focus is on the period of peak 24-nt phasiRNA production in 1.5 (mid-prophase 1) to 2.0 mm (end of meiosis 1) to 2.5 mm (tetrad stage) anthers.

6. All tapetal cells would become binucleate at end of meiosis. Do slightly larger anthers (2.5-3.5mm) in *dcl5-1* mutant also have fewer binucleate tapetal cells? This data would clarify whether tapetal development delayed or arrested.

You make a valid point, however, the *dcl5-1* anthers are failing by 3 to 4 mm, and it would

be difficult to distinguish a serious delay from anther failure if some tapetal cells remain mononucleate.

7. Line 76-78. “we conclude that the 24-nt phasiRNA pathway is localized in tapetal cells” might be overstated if the precursor of 24-nt phasiRNA are considered to be part of the pathway. Since according to Zhai et al. (2015) 24-PHAS are present in tapetal cells and in meiocytes.

Good point, except that the *in situ* hybridization method suffers from the artifact of probe adhering to callose. In our work in bioRxiv (cited above) using single cell RNA-seq of staged PMC and meiocytes, few 24-nt PHAS precursors were detected. Without precursors, the PMC and meiocytes cannot generate 24-nt phasiRNAs. This leaves open the possibility of transfer of 24-nt phasiRNAs from the tapetum to the germinal cells. The route is unknown, and if the callose coat is intact, no materials dissolved in aqueous media should be able to move into the germinal cells. Looking into the possibility of transfer awaits testing – in a future study – for the physical presence of diverse 24-nt phasiRNAs in purified germinal cells and analysis of the callose for potential channels. We estimate that this will require collecting ~5000 such cells free from tapetal contamination (a common occurrence in all of the manual collecting methods such as squeezing out the column of PMC and meiocytes from an anther lobe). In our protoplast collection method, collecting 5000 purified cells would require approximately 6 months and is far beyond the scope of the current study. One at a time, individual 24-nt phasiRNA products (of >3000 known) could be tested by higher resolution *in situ* detection methods; beyond testing a few such 24-nt phasiRNAs, this method is impractical for describing the entire suite of 24-nt phasiRNAs that might exist in PMC or meiocytes. Implementation of advanced microscopy techniques to examine the callose coat are in the planning stages at present. All of these feasible experiments are beyond the scope of the current study.

Minor points

1. Line 16: miRNA needs an abbreviation.

Fixed, thank you.

2. Fig S1 does not contain mRNA abundance data of dcl5 alleles.

Fixed, thank you. We had inadvertently duplicated Figure S7.

3. Fig S4, dcl5-1 mutant meiocytes have more organelle? what are those tiny DAPI-stained dots?

To our eyes, there is no obvious difference. The small dots are perhaps mitochondria, which would explain the DAPI staining.

4. Line 76, (4)?

This was a reference error, fixed.

5. Fig S7, dcl5-1 mutant under permissive and restrictive temperature showed higher expression of non-phasi 24-nt sRNAs.?? What they are? Why it is not seen in the dcl5-1 mutant at the 1.5 or 2 mm anthers?

Without the 24-nt phasiRNAs present, there is, in effect deeper sequencing of all other classes of small RNA, because sequencing reads are essentially the same. The data are not normalized to some invariant small RNA, and instead are just normalized to total sequences.

Therefore, more non-phasiRNA 24-nt small RNAs are sequenced. This sort of “enrichment” effect has been described for small RNA mutants going back over a decade, for example in *rdi2* mutants that lack most 24-nt sRNAs, and therefore appear to be enriched for miRNAs. The variation in the permissive and restrictive *dcl5-1* plants likely resulted from experimental variation, as the plants and libraries were from a different experiment to the rest; a note to this effect has been added to the figure legend.

6. Line 40-47: I suppose that *dcl5-2*, *dcl5-3* and *dcl5-4* plants are sterile, based on descriptions in lines 84-86. The authors should mention whether these three alleles are sterile or fertile with data shown in supplemental figures.

Yes, under field conditions, all were sterile, and this point is now emphasized. The phenotypes were the same. Because the controlled environment experiments are very expensive (multiple chamber and greenhouse rental), we used only the *dcl5-1* allele for the more extensive analyses.

7. Figure A-E, G-K in Figure S4 should be enlarged for easier reading.

Fixed, thank you.

8. Both *dcl5-2* and *dcl5-3* allele have normal DCL5 expression but without 24nt phasiRNAs (Figure S1). It would be an interesting to examine whether any phenotypic difference can be observed in tapetal cells and meiosis.

We agree that it will be interesting to check the tapetum and meiosis in all our *dcl5* alleles. *dcl5-2* and *dcl5-3* are small, in-frame deletions, while *dcl5-1* and *dcl5-4* are frameshift mutations. Because 24-nt phasiRNA are so greatly reduced in all four cases, it seems likely that the endoribonuclease activity is lost, despite the expression level. Furthermore, all *dcl5* alleles were male sterile under field conditions. We focused our work on *dcl5-1* as the representative allele, as imaging and count of binucleate cells took several months. Since the CRISPR system is still active in plants for which we have not segregated away, we are still generating even more alleles of *dcl5* mutants, with another five or six identified in recent months; however, we have yet to any characterization or clean up work on these, so this will take several more years – a resource for the future.

9. Fig S5 is incomplete.

Fixed, thank you.

Reviewer #2 (Remarks to the Author):

The manuscript, entitled “Dicer-like 5 deficiency confers temperature-sensitive male sterility in maize” (Zhang, et al) reports novel findings regarding the role that DICER-LIKE 5 plays in maintaining maize male fertility. DICER-LIKE (DCL) proteins are essential for biogenesis of small regulatory RNAs in angiosperms. Unlike other members (DCL1, DCL2, DCL3, and DCL4) that are conserved across all flowering plants, DCL5 was recently found monocot-specific. In this manuscript, authors created maize Dcl5 mutations using CRISPR-Cas9 gene editing technique, characterized maize *dcl5* mutants under different temperature-controlled growth condition and discovered that maize DCL5 is essential for robust male fertility. On the molecular level, this paper suggests that DCL5 corresponds to biogenesis of 24-nt

phasiRNAs, and the phasiRNA pathway is believed localized in tapetal cells and plays important roles in plant reproductive process. Findings from this study has not only advanced our understanding of DCL5's role in maintaining a robust male fertility and responding to the temperature changes, but also has significance in agricultural practice. I only have some minor suggestion and comments.

We appreciate the positive comments on our work.

1. By looking at the Figure 2, which seems to me that *dcl5-1* mutant anthers are not only developing less binucleate tapetal cells, but also abnormal in cell number and size. I am not sure if this is just from the particular section for imaging. From Figure 2B, it is clear that *dcl5-1* has smaller number but larger sized tapetal cells, middle layer cells, and endothelial layer cells—almost all the anther wall layers have larger sized and smaller numbered cells. Please clarify whether this applies to all *dcl5-1* anthers growing in 28°/22°C or in the field. Are there any statistic data regarding this abnormality.

Actually the cell size and number are the same in *dcl5-1* compared to fertile siblings, and this conclusion derives from observations as reported in Fig. S5: there are normal cell layers and morphology in *dcl5-1*. The TEM photographs in the original Fig.2 A/B were chosen to illustrate bi-nucleate and mono-nucleate tapetal cells; they were not chosen to address cell size or number in *dcl5* mutants. Therefore the extra TEM pictures which were originally in Fig.S7 have been used in the new version of Fig. 2 A and B, to better visualize the cells.

2. This is just following my first question. If *dcl5-1* mutants develop fewer but larger anther wall cells, it suggests a role before meiosis and likely on anther cell differentiation. In this case, it would make sense that the accumulation of 24nt phasiRNA in meiocytes and anthers (mainly tapetal cells) are parallel processes. The reduced number of binucleate tapetal cells in *dcl5* mutant could be associated with less dynamic cell division. Are there any other changes that the authors have been notified such as changes of gene expression of cyclin or cyclin dependent kinase genes?

By our analysis there is no cell size/number phenotype in *dcl5*, therefore it is unlikely that *Dcl5* transcripts have an earlier role. Furthermore, we have no evidence suggesting an earlier role for DCL5 protein. Transcripts are first detected shortly before the biogenesis of 24-nt phasiRNA starts. In our differential expression gene analysis with the RNA-seq data, we found no evidence of cyclin or cyclin dependent kinase genes standing out as impacted in the mutant anthers.

3. Since this paper suggests a correlation between the temperature sensitivity of DCL5 to the agricultural belt of North America, a couple more sentences to introduce the origin of maize Hi-II line would be helpful for readers. In addition, it may be helpful to mention that it may vary among maize inbreds, especially for those with tropical origin, such as CML228.

The Hi-II line is a cross between a Minnesota early line (A188) that confers regeneration capacity to excised embryos and B73, an Iowa Stiff Stalk Synthetic derivative. B73 and its derivatives were one of the major parents of hybrid corn sold in the US for 20 years; it is an exceptionally “well adapted” line with high yield potential. Both St. Paul Minnesota and Ames

Iowa where the lines originated experience high summer heat, both consistent high temperature and heat waves. While these particular points are perhaps not so relevant to the readers, we certainly appreciate the reviewer's point that there are likely background effects and the *dcl5* mutant phenotype may vary across inbreds. As suggested, we have added a sentence to this effect in the discussion.

Reviewer #3 (Remarks to the Author):

This study shows that DCL5 (which is DCL3b) is required for anther wall developments and this *dcl5* deficiency shows the temperature-sensitive male sterility through the decrease of 24-nt phasiRNAs in maize. However, it is already known that rice DCL3b, which is DCL5 in this paper, are required for 24-nt reproductive phasiRNA production in rice (Song et al., 2012). Furthermore, there are no correlation between 24-nt phasiRNAs expression and temperature-sensitive male sterility of *dcl5* mutants under restrictive and permissive temperature. It remains unknown the mechanism of temperature-sensitive male sterility via 24nt-phasiRNAs and the function of 24 reproductive phasiRNAs processed by DCL5. I would recommend against its publication in Nature Communications.

In our opinion, our work is an advance over the Song et al. (2012) paper, as in that work, they utilized an RNAi construct which reduced but did not eliminate *Dcl5* transcripts. There was no developmental phenotype observed, and it was impossible to assess redundancy between DCL3 and DCL5. Between our current manuscript and the companion paper, it's clear that DCL5 has distinct, completely non-redundant functions with DCL3, and thus is entirely deserving a separate gene name. The a/b designation should be reserved for redundant, lineage-specific gene duplicates. The DCL3 gene duplication pre-dates the emergence of the grasses, and subfunctionalization of DCL3 and DCL5 has definitely occurred as documented in our manuscript.

Major comments

1) DCL5 allele mutants

There are no data of mendelian inheritance of "EACH" *dcl5* allele (page 2 Line37).

Please show these results including under 28/22°C, permissive and restrictive temperature.

Genetic crosses in the greenhouse and the field established that each of the alleles exhibits male sterility and Mendelian inheritance; additional Table S1 and Fig. S1 have been added to the manuscript. Table S1 shows the Mendelian inheritance of *dcl5* alleles.

For the particular case of *dcl5-1*, the focus of most studies in our manuscript, a T3 family generated by self-pollination of a heterozygote (no phenotype) showed progeny segregation of 3:1 fertile:sterile and genetic segregation of 1:2:1 as expected.. Most *dcl5-1* families have been propagated by crossing homozygous, sterile individuals by heterozygous siblings, routinely generating the expected 1:1 phenotypic and genotypic segregation ratio. Though it was mainly focused on *dcl5-1* families; fewer families were tested for the other alleles. The newly added Fig. S1. provides evidence that male sterility is perfectly co-segregated with the homozygous mutants for all alleles, which suggested the male sterility phenotype reflects null function of *dcl5*

gene. The male sterility phenotype of *dcl5-1* in the field, *dcl5-2* and *dcl5-4* is also shown (Fig. S1. B, C and D). We do not have a publication quality photo of *dcl5-3*.

Please check whether there are other deletion or insertion regions in all mutants by whole genome sequences or other experiments, because there are 5 paralogues of DCLs in maize.

Good point and we've tried several analyses to pursue this. First, neither BLAST nor "target prediction" using both guide RNAs (gRNA) against the maize genomic sequence identified potential off-targets. Based on protein sequence homology, only *Dcl3* is highly similar to *Dcl5* while *Dcl1/2/4* are quite distinctive (new figure, Fig.S4. A-D). Furthermore, the homologous coding sequences are even less conserved.

Therefore, *Dcl3* was compared to *Dcl5* in greater detail. The first gRNA is poorly matched at the 3'-end to *Dcl3* and the second gRNA lacks the PAM site (Fig. S4, E). The gRNAs were designed to distinguish *Dcl5* from *Dcl3*. We also queried the RNA-seq reads for *Dcl3* and *Dcl5* including all the *dcl5* mutants and their fertile siblings. The wildtype (non-edited) gRNA homologous sequences in *Dcl5* are only found in the control libraries or the non-edited loci in heterozygous mutants, confirming that the gRNAs targeted *dcl5*. In contrast, all *Dcl3* reads from this analysis were mostly wildtype in all libraries (deviations are likely sequencing errors at individual bases and occurred mainly outside the gRNA region), and no off-target editing could be confirmed at those loci.

We have four independent *dcl5* null alleles in which 24-nt phasiRNAs are very greatly reduced, and the homozygous mutants also exhibit male sterility. We also noted a "dwarf" trait after selfing of two initial families (actually derived from the same tissue culture line), but this trait was not linked to the *dcl5* locus. We ascribed it to a mutation during tissue culture rather than an offsite effect, because poor growth "mutants" or "epimutants" are common after maize tissue culture, and two sister lines had the same defect.

Thus far, introgression of mutant alleles (currently at the BC1 to BC3 stage) has not resulted in a change in phenotype; introgression is the standard method for eliminating unlinked off-target mutations. For the particular case of *dcl5-1*, we are at BC3 (93.75%) in our standard line W23; male-sterility occurs in homozygous mutants only and we have not observed any other phenotype.

Please show the expression of Dcl5 in all mutants. I couldn't check the results of Fig S1.

Fixed; as noted in our response to reviewer 1, minor comment #2, this was a copy-paste error duplicating a figure. We apologize!

2) Female effect in *dcl5*

How about female in *dcl5* mutants?

The *dcl5* mutants were all female fertile, as we were able to use them as the pollen recipient in each case, with no apparent reduction in fertility. We did not examine them at a cytological level as there was no evidence of a fertility defect.

3) Line49-63

Please explain the mono- and bi-nucleotide more detail with references.

Tapetal cells in nearly all angiosperms achieve binucleate status during meiosis (see Walbot and Egger, 2016 and references therein). This was described as early as the 1950s (Scarascia 1953, Davis 1966, Buss and Lersten 1975, Franceschi and Horner 1979, Castillo 1988, Horner 1993). In Horner's 1993 paper, the author found the tapetum consisted of a mixture of mono- and bi-nucleated cells during microsporogenesis.

There is no current explanation for this unusual cytological feature. Many plant cells achieve polyploidy, which is argued to increase transcript output within a single larger cell and to be a more efficient use of resources in terminally differentiated cells. Tapetal cells during meiosis are not terminally differentiated, because they will acquire new properties after meiosis and exhibit new roles in supporting the microgametophytes. Why binucleate status rather than tetraploidy? No one has proposed a testable explanation for this unusual phenotype.

4) *Dcl5* expression and DCL5 protein expression (line 69-79)

Please show DCL5 and DCL3a expression with in situ hybridization and show DCL5 and DCL3a protein localization in anther, because DCL3 is also important for 24-nt reproduction in dicots (Xia et al paper).

This is beyond the scope of our study. There are no good antibodies for plant Dicer genes; plus, the genomic sequence of *Dcl5* is 50 kb, so making and analyzing an epitope tagged version would take several years.

5) 24-PHAS expression

Please explain the difference of 24-PHAS expression in *dcl5-4* and other three alleles (FigS8), though the expression of 24-nt phasiRNAs are decreased in all alleles (Fig3 and FigS7).

We do not have an explanation for the slight variation.

6) 21-nt phasiRNAs/21PHAS

Why were 21-nt phasiRNAs increased in all *dcl5* mutants (FigS7) ?

How about 21PHAS expression in mutants?

In normal anthers the 21-nt phasiRNAs peak at 0.4 mm but could continue to be produced through 1.0 mm (precursors are still present) and the 21-nt phasiRNAs are present at steadily declining low levels at later stages. If the *dcl5-1* mutant anthers are delayed in development, it is plausible that there would be a delay in the decay or destruction of the 21-nt phasiRNAs, which are present throughout the soma including in the tapetal cells. This is just a speculation, however.

As mentioned above in response to a similar question by reviewer 1 (minor point #5), the increase of 21-nt phasiRNA in *dcl5* alleles is likely just a relative effect of removing the abundant 24-nt phasiRNAs but sequencing to the same depth.

Maybe another way to answer the question is that our differential expression analysis showed no obvious up-regulation of 21-PHAS precursors, mainly because 21-PHAS are exclusively expressed in the earlier anthers (before 1.0 mm), while we are looking at 2.0 mm anthers. Also, *Dcl5* is very weakly expressed before the 1.5 mm anther stage and therefore, it's unlikely to impact the 21-nt phasiRNA pathway.

7) There are no significant differences of “24nt phasiRNAs expression (also 24-PHAS)” under restrictive and permissive temperature (Fig3 and FigS8). Authors concluded that 24-nt phasiRNAs are dispensable at cool growing temperature for maize and may allow alternative pathways (line111,112, 113150). Please show the evidence. For example, the authors performed RNA-seq using WT and *dcl5* under restrictive and permissive temperature. How about changing of genes of alternative pathways independent of 24nt phasiRNAs to support the authors interpretation?

Our point that the 24-nt phasiRNAs are dispensable at cooler temperatures is simply a statement of fact, not an interpretation. In other words, we grew the *dcl5-1* plants at lower temperatures, there were few if any 24-nt phasiRNAs, and the plants are fertile – *ipso facto*, the pathway is dispensable. As to whether there are other pathways that are active or whether the impact of the absence of the 24-nt phasiRNAs has a direct or indirect effect – this will require additional experimentation and is the topic of our future work.

8) Targets of 24-nt phasiRNAs

The authors concluded that characterization of downstream transcription, target, was uninformative (line 92). Please show the results. Because it seems that premeiotic phasiRNAs (21nt?) are used for searching target, not meiotic phasiRNAs in Zhei et al (2015) paper, which are referenced (line 93).

We have no evidence that either 21- or 24-nt phasiRNAs target mRNAs. Basically, at high levels of pairing, there are few targets, and at very relaxed levels, they find many targets. In either case, it has not been possible to validate cleavage, perhaps because of the small number of cells in which they act. We have recently published a pair of papers in *New Phytologist* that describe these analyses (DOI: 10.1111/nph.15181 and 10.1111/nph.15349). Given that all known plant small RNAs interact with some other RNA or with DNA (in the case of the 24-nt small RNAs involved in gene silencing), they surely have some sort of target, but what are these targets, and what activities do they direct to these targets? This is central to the continuing “mystery” of their function(s).

9) It is recommended to fit this manuscript to Nature Communications form.

The manuscript has been revised to conform to the journal requirements.

Reviewers' comments:

Reviewer #1 (Remarks to the Author):

This manuscript, entitled "Dicer-like 5 deficiency confers temperature-sensitive male sterility in maize" reports the involvement of DCL5 in 24-nt phasiRNA biogenesis in maize and shows that DCL5 and 24-nt phasiRNAs play vital roles in tapetal functions at high temperatures. In general, the findings are very interesting. However, I still have some comments.

1.If DCL5 and 24-nt phasiRNAs are only indispensable under higher temperature, but not required for anther development under cooler temperatures, one would expect to see increased expression of DCL5 in the wild-type anthers under higher temperature. Results shown in Fig. S2 are from *dcl5* mutant anthers under permissive and restrictive temperature, which does not explain whether DCL5 and 24-nt phasiRNA respond to heat treatment in the wild-type.

2.The impacts of DCL5 on the tapetum are obvious, and the tapetum is the primary site of 24-nt biogenesis. It has previously been shown that tapetal cells play several important roles during male reproduction, such as supporting meiocytes, secreting enzymes to digest callose walls, mediating microscope development, and depositing components for pollen coats. Since the tapetum layer has multiple functions, it is necessary to determine what stage of gametogenesis is affected. Only showing abnormal tapetum morphology is a bit disappointing. Moreover, for example, if callose degradation is impaired in the *dcl5* mutant, it would support the authors' speculation that 24-nt phasiRNAs are essential for tapetal redifferentiation for biosynthesis and secretion of materials that support haploid microgametophytes (lines 186 and 187).

3.Were tapetum cells of the *dcl5* mutant binucleate under permissive temperate?

4.In the rice *tip2* mutant in which expression of DCL5 and production of 24-nt phasiRNA are severely affected (PLoS Genet, 14(2):e1007238), 5-7% of male meiosis showed segregation defects. Is it possible that the maize *dcl5* mutant exhibits mild meiotic abnormality?

5.In the comparison of Figs. 2C and 2D, nuclei in Fig. 2C (*dcl5* mutant?) are larger and have a higher percentage of binucleate tapetal cells than nuclei shown in Fig 2D (wild-type). This is not consistent with what is stated in the main text. Please clarify.

6.Line 445: In the figure legend, the orange boxes mark "two" binucleate tapetal cells. However, I only see one tapetal cell in the orange box.

7.Although the authors state that "In extensive confocal microscopic examination of *dcl5-1* anthers, we observed no significant or consistent differences in cell number or volume compared to fertile siblings" (lines 91-92). However, by examining Fig. S8, I feel it is very difficult to make such a conclusion. Statistical analysis is required to support this point.

8.The scale bar in Fig. 4D-4I may be incorrect.

Reviewer #3 (Remarks to the Author):

To Authors,

This revise is getting better than before, but there were parts that the authors do not answer to my questions. Though authors claim that 24-nt phasiRNAs are dispensable at cooler temperatures, more experimental evidence should be shown towards this statement. Because there is no correlation between 24-nt phasiRNAs expression and temperature-sensitive male sterility of *dcl5* mutants under restrictive and permissive temperature. If this revise is included in the "24-nt reproductive phasiRNAs are broadly present in angiosperm", not back to back, the data and conclusion are reasonable to Nature Communications.

1) RNA/small RNA-sequencing using "WT or control" under permissive and restrictive temperature conditions seem to be important to reveal the expression of DCL5, phasiRNAs and 24-PHAS under

high and low conditions.

2) Mendelian inheritance of *dcl5-1* (Sup table1)

Please show mendelian inheritance analysis using backcross segregants (N>50), because the authors produced the BC1 to 3 of *dcl5-1* (Sup table 1).

3) Sup figure 10

In this revise version, the abundance pattern in *dcl5* mutants in SupFig10 is different from that of previous same figure (Sup fig8 in previous version), especially *dcl5-1* permissive and restrictive seems to be opposite. Which figure is true?

Reviewers' comments:

Reviewer #1 (Remarks to the Author):

This manuscript, entitled "Dicer-like 5 deficiency confers temperature-sensitive male sterility in maize" reports the involvement of DCL5 in 24-nt phasiRNA biogenesis in maize and shows that DCL5 and 24-nt phasiRNAs play vital roles in tapetal functions at high temperatures. In general, the findings are very interesting. However, I still have some comments.

1. If DCL5 and 24-nt phasiRNAs are only indispensable under higher temperature, but not required for anther development under cooler temperatures, one would expect to see increased expression of DCL5 in the wild-type anthers under higher temperature. Results shown in Fig. S2 are from *dcl5* mutant anthers under permissive and restrictive temperature, which does not explain whether DCL5 and 24-nt phasiRNA respond to heat treatment in the wild-type.

Response: The "restrictive" temperature is the optimal temperature for corn growth but it is not a heat treatment; we compared the *Dcl5* and *24-PHAS* abundance in WT (W23) under restrictive and permissive temperature, but we found no significant differences (Fig.S2 and Fig.S12, W23 under permissive and restrictive temperature). Furthermore, we compared the sum of 24-nt phasiRNA abundance under these two conditions, which showed no significant difference either (Fig. 3, W23 under permissive and restrictive temperature). Considering this, we hypothesize that the lower, permissive temperature restores fertility by influencing anther growth and development, somehow obviating the need for the 24-nt phasiRNAs, rather than directly impacting the 24-nt phasiRNA pathway.

2. The impacts of DCL5 on the tapetum are obvious, and the tapetum is the primary site of 24-nt biogenesis. It has previously been shown that tapetal cells play several important roles during male reproduction, such as supporting meiocytes, secreting enzymes to digest callose walls, mediating microscope development, and depositing components for pollen coats. Since the tapetum layer has multiple functions, it is necessary to determine what stage of gametogenesis is affected. Only showing abnormal tapetum morphology is a bit disappointing. Moreover, for example, if callose degradation is impaired in the *dcl5* mutant, it would support the authors' speculation that 24-nt phasiRNAs are essential for tapetal redifferentiation for biosynthesis and secretion of materials that support haploid microgametophytes (lines 186 and 187).

Response: Again, we appreciate the question. We showed that meiosis can be completed and some exine forms (Fig. S7M, N) while the middle layer and tapetum are delayed or altered (Fig. S10). Moreover, the tapetal alteration starts as early as the 1.5 mm anther stage during early prophase I (Fig. 2) after the remodeling of the callose coat on the PMC, which finishes by ~1.2 mm anther stage. Uninucleate microspores are released after meiosis, indicating that the tapetal secretion of beta-glucanase is sufficient at this stage. The later defects in gametophyte development and gametogenesis are therefore secondary, which may explain why portions of tassels have fertile anthers under the permissive conditions. Overall, our cytological data are consistent with a function of *Dcl5* and 24-nt phasiRNAs to determine tapetal cell redifferentiation in 1.5 – 2.5 mm anthers, and male fertility is dependent on this cellular state.

3. Were tapetum cells of the *dcl5* mutant binucleate under permissive temperatures?

Response: The percentage of binucleated tapetal cells at 1.5 and 2.5 mm in *dcl5-1* anthers was similar to their normal siblings under the permissive conditions (Fig. S15). This is reflected in main text line 153-155.

4. In the rice *tip2* mutant in which expression of DCL5 and production of 24-nt phasiRNA are severely affected (PLoS Genet, 14(2):e1007238), 5-7% of male meiosis showed segregation defects. Is it possible that the maize *dcl5* mutant exhibits mild meiotic abnormalit

Response: In our reading of that paper, the anther wall cells in the rice *tip2* mutants seem less differentiated and therefore more severely affected than our maize *dcl5* mutants. TIP2 is a transcription factor regulating many aspects of tapetal status, not just 24-nt phasiRNA biogenesis. TIP2 is the ortholog of maize Ms23, a master regulator of tapetal development (Nan et al. 2017), and we have previously shown that in *ms23* the 24-nt phasiRNAs are missing (Zhai et al. 2015). MS23 is required for expression of *Dcl5* in maize (Nan et al. 2017), and this may also be true for TIP2 in rice. Given the role of *Ms23*, we infer that the loss of TIP2 in rice likely affects more tapetal pathways and could influence more meiosis-specific processes, starting from an earlier stage than the loss of DCL5 and 24-nt phasiRNAs. We wonder if male sterility in *tip2* could be recovered in permissive conditions. It appears to us that the rice *tip2* and maize *dcl5* mutants are not entirely comparable, but more work needs to be done, for example, determining comparable growth conditions.

5. In the comparison of Figs. 2C and 2D, nuclei in Fig. 2C (*dcl5* mutant?) are larger and have a higher percentage of binucleate tapetal cells than nuclei shown in Fig 2D (wild-type). This is not consistent with what is stated in the main text. Please clarify.

Response: Thank you for pointing out this issue in the legend of Fig. 2C and Fig. 2D; these are now corrected.

6. Line 445: In the figure legend, the orange boxes mark “two” binucleate tapetal cells. However, I only see one tapetal cell in the orange box.

Response: We have enlarged the bi-nucleated tapetal cells in Fig. S8E-G.

7. Although the authors state that “In extensive confocal microscopic examination of *dcl5-1* anthers, we observed no significant or consistent differences in cell number or volume compared to fertile siblings” (lines 91-92). However, by examining Fig. S8, I feel it is very difficult to make such a conclusion. Statistical analysis is required to support this point.

We thank for the reviewer's concern and understand it will be great to set up a standard to characterize the cell number and sizes of tapetum. While it is not very much relevant to the scope of this work, therefore we didn't include it here. Hopefully we can achieve this goal and clarify it in a near future.

8. The scale bar in Fig. 4D-4I may be incorrect.

Response: Thank you, we have corrected the bar size.

Reviewer #3 (Remarks to the Author):

This revision is getting better than before, but there were parts that the authors do not answer to my questions. Though authors claim that 24-nt phasiRNAs are dispensable at cooler temperatures, more experimental evidence should be shown towards this statement. Because there is no correlation between 24-nt phasiRNAs expression and temperature-sensitive male sterility of *dcl5* mutants under restrictive and permissive temperature. If this revise is included in the “24-nt reproductive phasiRNAs are broadly present in angiosperm”, not back to back, the data and conclusion are reasonable to Nature Communications.

1) RNA/small RNA-sequencing using “WT or control” under permissive and restrictive temperature conditions seem to be important to reveal the expression of *DCL5*, phasiRNAs and 24-PHAS under high and low conditions.

Response: To address this reviewer's concern, in newly-added data, we compared *Dcl5* and 24-PHAS abundance in WT (W23 inbred) under restrictive and permissive temperature. No significant difference was observed (Fig. S2 and Fig. S12, W23 under permissive and restrictive temperatures). Furthermore, we compared the sum of 24-nt phasiRNA abundances under these two conditions, which also showed no significant difference (Fig. 3, W23 under permissive and restrictive temperature). Considering these data, we conclude that the permissive temperature allows fertility recovery by impacting anther growth and development to obviate or bypass the required functions of the 24-nt phasiRNAs, rather than directly impacting the 24-nt phasiRNA pathway.

2) Mendelian inheritance of *dcl5-1* (Sup table1)

Please show mendelian inheritance analysis using backcross segregants (N>50), because the authors produced the BC1 to 3 of *dcl5-1* (Sup table 1).

Response: We took an additional field season to address this reviewer's concern, hence the delay of nearly a year since our previous revision. More genetic data was added to this study including the analysis of more segregants, and perhaps most significantly a newly-obtained Mutator (transposon) allele (*dcl5-mu03*), distinct from the previously-described CRISPR alleles. We can now demonstrate the Mendelian segregation of the phenotype of these alleles. These changes are reflected in the main text in lines 59-60, also in Table S1, Fig. S5, Fig. S11, and Fig. S13.

3) Sup figure 10

In this revised version, the abundance pattern in *dcl5* mutants in Sup fig 10 is different from that of previous same figure (Sup fig 8 in previous version), especially *dcl5-1* permissive and restrictive seems to be opposite. Which figure is true?

Response: We appreciate the carefulness of reviewer and we have corrected this, with the confirmed, current results now in Fig. S12.

REVIEWERS' COMMENTS:

Reviewer #3 (Remarks to the Author):

Dear Authors,

Compared to the manuscript a year ago, a lot of supplementary data have been added, making the revise version an attractive article.

1) Quantitative data for fertility/sterility (Fig4, SupFig16 and Sup Table1)

I'm interested in the swap analysis between the high and low temperature of Supplementary Fig 16. But it is unclear to distinguish which tassels are fertile or sterile, because the anthers in most tassels were detected in these figures photo. I recommend a quantitative indicator to show the fertility rate using pollen fertility rate or seed fertility rate etc, NOT PHOTO. I recommend the quantitative fertility data for swap analysis is added in the main Fig. 4, because these data are important.

In Fig4, I detected (D, E, H and I) are fertile and (G) is sterile. However, it is unclear in (F) photo.

in Sup table1, there is no quantitative data for fertility/sterility with segregation of siblings. For example, how about seeds fertility rate? I don't know whether seeds fertility of "+/+ " and "+/- " siblings are 100%, and those of "-/-" are 0%.

2) Fig. 2 E and Supplementary Fig 15

The numbers of binucleated tapetal cells in DCL5/DCL5, which showed fertility is low, around 15% under permissive temperature in Supplementary Fig 15.

This rate (15% of DCL5/DCL5) is less than the rates (28%) of dcl5-1/dcl5-1, which shows sterile (in Fig2 E). Is it correct?

3) Supplementary Fig 12

It seems that the total expression patterns of dcl5-1 restrictive are higher than those of dcl5-1 permissive (log2 of abundance). But it seems that box-plots (right) are opposite between dcl5-1 restrictive and dcl5-1 permissive.

4) Supplementary Fig 9 C and D

The pictures of C and D were not clear, please change the photos (if you could do)

Response to reviewer #3:

Thank you to reviewer #3 for the helpful comments! Here are our responses to each of your comments, in *blue italics*, as follows:

1) Quantitative data for fertility/sterility (Fig4, SupFig16 and Sup Table1)

I'm interested in the swap analysis between the high and low temperature of Supplementary Fig 16. But it is unclear to distinguish which tassels are fertile or sterile, because the anthers in most tassels were detected in these figures photo. I recommend a quantitative indicator to show the fertility rate using pollen fertility rate or seed fertility rate etc, NOT PHOTO. I recommend the quantitative fertility data for swap analysis is added in the main Fig. 4, because these data are important.

In Fig4, I detected (D, E, H and I) are fertile and (G) is sterile. However, it is unclear in (F) photo.

To better support our data and conclusions, we have now explained our previous method and added an additional, quantitative analysis for determination of male fertility in this study. We added the following details to the manuscript:

In brief, anther exertion on male fertile tassel take approximately seven days: Anthers emerge from the upper florets of the center of main spike on day 1; day 2 will include anthers from the lower florets in the center of the main spike and additional upper florets from the main spike (above and below the middle zone); day 3 will include lower floret spikes from the zones that stated on day 2, plus the upper florets at the base of the main spike and the middle of the primary branches; day 4 will include lower florets from the spikelets involved on day 3, plus upper florets from more regions on the primary branches; day 5 will include the lower florets from spikelets involved on day 4; day 6 will include the upper florets on the secondary branches, and day 7 will include the lower florets of the secondary branches 24. Based on the maize anther emergence, tassel at similar stages in anther emergence were chosen for photographs and quantification.

Tassel images obtained with cell phones with digital cameras. Images were then subjected to a custom machine learning workflow based on PlantCV (<https://plantcv.danforthcenter.org>). In brief, Naïve Bayes approach was applied

and three classes (anther, other tassel parts, and background) of labeled pixels were used to train in the PlantCV environment. By using these three classifiers, anthers and other parts of tassel were highlighted in red and green. Then, area of anther and tassel from each tassel was summed and used to calculate anther/tassel area ratio, as an indicator for male fertility.

To better demonstrate the male fertility in each tassel images in Fig.4 and Supplementary Figure 16, anther/tassel area ratio is used. With this method, we analyzed Fig.4, and a Supplementary Figure 12 is added to demonstrate the male fertility; tassel images in the temperature swap experiment were analyzed and the new results have been added to Supplementary Figure 17.

in Sup table1, there is no quantitative data for fertility/sterility with segregation of siblings. For example, how about seeds fertility rate? I don't know whether seeds fertility of "+/+” and “+/-“ siblings are 100%, and those of “-/-“ are 0%.

We applied constant restrictive temperature conditions (indicated in footnote b, c, d, below the table) to grow the populations listed in Supplementary Table 1, therefore homozygous dcl5 plants were male sterile, as shown in Fig.1 and Supplementary Figure 5; the heterozygous and wildtype siblings were fully fertile and in contrast to the homozygous plants. We have added this clarification in the footnote a in Supplementary Table 1.

2) Fig. 2 E and Supplementary Fig 15

The numbers of binucleated tapetal cells in DCL5/DCL5, which showed fertility is low, around 15% under permissive temperature in Supplementary Fig 15.

This rate (15% of DCL5/DCL5) is less than the rates (28%) of dcl5-1/dcl5-1, which shows sterile (in Fig2 E). Is it correct?

It is correct that percent of the same genotype at same stages is higher in Fig. 2E for few reasons: 1) The percent was analyzed based on 3D reconstruction images in Fig. 2E, while that in Supplementary Figure 15 was based on 2D confocal images, which

made the numbers not quite comparable; 2) The cooler temperature of permissive conditions may have reduced by half the cell endomitosis.

3) Supplementary Fig 12

It seems that the total expression patterns of *dcl5-1* restrictive are higher than those of *dcl5-1* permissive (log2 of abundance). But it seems that box-plots (right) are opposite between *dcl5-1* restrictive and *dcl5-1* permissive.

*Thank you. We double-checked the data and found that *dcl5-1* restrictive and *dcl5-1* permissive were switched and the box plot was right. We have fixed this mistake in the revision.*

4) Supplementary Fig 9 C and D

The pictures of C and D were not clear, please change the photos (if you could do)

Thank you for pointing it out! Unfortunately, we had no better versions of the images that we can substitute in Supplementary Figure 9C, D. It was very difficult to improve the image quality due to the limitation of technology in anthers larger than 3.0 mm. Even though, the image quality was good enough for manually counting and the differences between mutant and their siblings were demonstrated in Supplementary Figure 9E. We believe the technology will be improved in the future, but this is the best we can do for now.